# A Critical Review for Synergic Kinetics and Strategies for Enhanced Photopolymerizations for 3D-Printing and Additive Manufacturing

**DOI:** 10.3390/polym13142325

**Published:** 2021-07-15

**Authors:** Jui-Teng Lin, Jacques Lalevee, Da-Chun Cheng

**Affiliations:** 1New Vision Inc., 10F, No. 55, Sect.3, Xinbei Blvd, Xinzhuang, New Taipei City 242, Taiwan; 2CNRS, IS2M UMR 7361, Université de Haute-Alsace, F-68100 Mulhouse, France; jacques.lalevee@uha.fr; 3Department of Biomedical Imaging and Radiological Science, China Medical University, Taichung 404, Taiwan

**Keywords:** 3D printings, additive manufacturing, polymerization kinetics, photoredox, monomer conversion, synergic effects, cationic, free radical, UV visible light

## Abstract

The synergic features and enhancing strategies for various photopolymerization systems are reviewed by kinetic schemes and the associated measurements. The important topics include (i) photo crosslinking of corneas for the treatment of corneal diseases using UVA-light (365 nm) light and riboflavin as the photosensitizer; (ii) synergic effects by a dual-function enhancer in a three-initiator system; (iii) synergic effects by a three-initiator C/B/A system, with electron-transfer and oxygen-mediated energy-transfer pathways; (iv) copper-complex (G1) photoredox catalyst in G1/Iod/NVK systems for free radical (FRP) and cationic photopolymerization (CP); (v) radical-mediated thiol-ene (TE) photopolymerizations; (vi) superbase photogenerator based-catalyzed thiol−acrylate Michael (TM) addition reaction; and the combined system of TE and TM using dual wavelength; (vii) dual-wavelength (UV and blue) controlled photopolymerization confinement (PC); (viii) dual-wavelength (UV and red) selectively controlled 3D printing; and (ix) three-wavelength selectively controlled in 3D printing and additive manufacturing (AM). With minimum mathematics, we present (for the first time) the synergic features and enhancing strategies for various systems of multi-components, initiators, monomers, and under one-, two-, and three-wavelength light. Therefore, this review provides not only the bridging between modeling and measurements, but also guidance for further experimental studies and new applications in 3D printings and additive manufacturing (AM), based on the innovative concepts (kinetics/schemes).

## 1. Introduction

The advantages of photopolymerization over the conventional thermal-initiated polymerization include the following: (i) Fast and controllable reaction rates, (ii) spatial and temporal control over the formation of the material, without the need for high temperatures or harsh conditions, and (iii) synergic effects and enhancement are available by using co-initiators or catalytic complexes [1,2]. Photopolymerizations using various lights with wavelengths in UV, visible, and near IR have been studied for both industrial and medical applications. A variety of photoresponsive materials such as conjugated polymers have been reported for additive manufacturing (AM) and recently for 3D and 4D bioprinting [3,4,5,6,7,8,9,10,11]. Both spatial and temporal controlled 3D processes were reported using single-and multiple-wavelength lights. For 3D photo printings, the key factors include polymerization depth, resolution precision, and speed, in which the monomer conversion efficacy could be improved by various strategies [12,13,14,15,16,17,18]. They include (i) a thiol–Michael/acrylate hybrid, epoxy/acrylate curable resins, thiol–acrylate/thiol–acetoacetate thermosets, and thiol–ene/epoxy-based polymers [12,13,14]; (ii)the use of novel materials as enhancers or co-initiators in both single and multiple components [15,16,17]; and (iii) two-stage polymerization under two wavelengths to eliminate the oxygen inhibition effects [17,18,19]. 

UV light (at 365 nm) has been commonly used in most type-I photoinitiators for the photopolymerization of (meth)acrylate monomers [1,2,3]. However, the UV wavelength suffers the disadvantages of being unsafe to skin and eyes, having a small penetration depth, and larger light scattering in tissues [1,2]. Camphorquinone (CQ), due to its good visible absorption properties, is the most common type-II free radical photopolymerization of (meth-) acrylates under blue light [15,20]. The classical diaryliodonium salts, such as diaryliodonium, suffer low solubility in monomers and the formation of side products due to the release of HF. To overcome this drawback, Kirschner et al. [15] recently reported a new counter anion-free and fluoride-free aryliodonium ylides (AY) to avoid the formation of HF and to enhance their solubility. They reported (CQ)/amine/AY as a new and efficient PI system for the polymerization of methacrylates under air and blue light (477 nm) irradiation. Kirschner et al. [15] also reported the chemical mechanisms involved in the presence of various AY and amines, which lead to additional reactions and initiating radicals for improved conversion efficacy. 

In comparison, near-infrared (NIR) light offers advantages of safer, less light diffusion and scattering, and deeper penetration into the materials. Thus, the curing of a thick and filled material can be potentially enhanced compared to curing with UV or visible light. However, the use of NIR photoinitiating systems such as cyanine is often associated with low reactivity and requires a high light intensity. Phthalocyanines, conjugated macrocycles, have been used as commercial pigments and dyes with a high molar absorptivity coefficient in the red and NIR wavelength of 650–810 nm. Efficient polymerization conversions using NIR photoinitiation by cyanine/iodonium salt couples are reported by Schmitz et al. [16]. Recently, Bonardi et al. [17] reported the first three-initiator system for high-performance NIR (785 nm) photopolymerization of thick methacrylates, in which a dual-function enhancer (phosphine) was used to prevent oxygen inhibition, and to regenerate the PS upon irradiation, in which a stable radical is coupled with the enhancer. The three-initiator system with a dye as a photosensitizer absorbing in the NIR range, an iodonium salt (as an initiator), and a phosphine (as a co-initiator) was reported, in which the phosphine is used to reduce oxygen inhibition (OIH) during the free radical polymerization of (meth)acrylate monomers [17,18]. 

Oxygen inhibition plays a critical role specially for optically thin polymers. Various strategies to reduce oxygen inhibition in photoinduced polymerization have been proposed such as: (i) Using a higher photoinitiator concentration; (ii) using a higher light dose or intensity; (iii) using co-initiators; (iv) the addition of oxygen scavengers; and (v) working in an inert environment [19]. Besides the above methods, chemical mechanisms were also reported, such as the thiol-ene and thiol-acrylate-Michael systems, which are insensitive to oxygen [12,13,14]. An additive enhancer-monomer was proposed to improve the curing (crosslink) efficacy by either reducing the oxygen-inhibition effect by stable-monomer or increasing the lifetime of radicals in clinical applications. Dual-wavelength (red and UV) photopolymerization was also reported, in which pre-irradiation of the red light eliminated the oxygen-inhibition effect and thus enhanced the conversion efficacy of the UV light [10,18,19]. 

An example of a blue and UV dual-wavelength system (without the red-light) for enhanced conversion by reducing the oxygen inhibition was reported by de Beer et al. [8] and van der Laan et al. [9], in which a blue (470 nm) and a UV (365 nm) light were used for the photopolymerization of methacrylate formulated with camphorquinone (CQ) and ethyl 4-(dimethylamino)benzoate (EDAB), where CQ is the blue-light active initiator (A), butyl nitrite (BN) is the UV-activated initiator (B), and EDAB is a co-initiator (or donor D). Lin et al. [20] reported the theoretical modeling for the above-described two-wavelength system.

An example of a two-wavelength (red and UV) system (without the blue-light) for 3D printing was reported by Childress et al. [10], in which a monomer of ethyl ether acrylate (DEGEEA) was mixed by zinc 2,9,16,23-tetra-tert-butyl-29H,31H-phthalocyanine (ZnTTP) as an initiator under a UV-light, where ZnTTP/DEGEEA has a distinct absorption peak at UV-365 nm and red-635 nm, respectively, and thus it can be independently excited by a UV and a red light, respectively. Lin et al. [21] reported the theoretical modeling for the above-described two-wavelength system. A novel strategy using three wavelengths of UV, blue, and red lights was recently theoretically proposed by Lin et al. [22] for future experimental studies.

Multicomponent photoinitiating systems using dye as visible light absorbing compounds have also attracted much attention for visible light curing [23,24,25,26,27]. These systems, often based on dye/iodonium salt/amine combinations, often have the great advantages of simultaneously generating initiating radicals and cations species [25]. Therefore, these latter systems can advantageously initiate both free radical and/or cationic polymerization processes. Three-component (Coum/NPG/Iod) photoinitiating systems for the free radical photopolymerization of (meth)acrylates using the new synthesized set of in silico developed coumarin derivatives were investigated by Abdallah et al. [25,26], which offers two distinct strategies: A photooxidation approach based on an iodonium salt and a photoreduction approach based on an amine under visible LED as source of irradiation. 

Recently, another three components of G1/Iod/NVK and G1/Iod/EPOXY photoinitiating systems were invested by Mokbel et al. [28,29] using a copper complex (G1), in which the co-initiators/additives Iod/NVK have dual functions: (i) Regeneration of the photoinitiator, and (ii) generation of extra radicals. The synergic effects lead to higher conversion of free radical polymerization (FRP) and cationic polymerization (CP). The kinetics of the copper complex photoredox catalyst including the roles of oxygen, thickness, and the optimal concentration for radical/cationic hybrid photopolymerization was reported by Lin et al. [30]. 

Table 1 summarizes various reported enhancing strategies for photopolymerization including one component (or monomer) and one-wavelength, two-component, and one-, two-, and three-wavelength [9,10,11,12,14,19,20,21] and three-component and one-wavelength systems [26,27,28,29,30,31]. We note that all these systems have been theoretically and experimentally studied, except the three-wavelength systems, which were recently proposed theoretically by Lin et al. [22]. 

This article will review, for the first time, the following kinetics and the synergic features of various systems (as summarized in Table 1), in which measured data are analyzed by modeling.

(1)Photo crosslinking of corneas for the treatment of corneal diseases using UVA-light (365 nm) light and riboflavin as the photosensitizer.(2)Synergic effects by a dual-function enhancer in a three-initiator system (one-monomer).(3)Synergic effects by a three-initiator C/B/A system, with electron-transfer and oxygen-mediated energy-transfer pathways for free radical (FRP) and cationic photopolymerization (CP).(4)Copper-complex (G1) photoredox catalyst in G1/Iod/NVK systems for FRP and CP.(5)Radical-mediated thiol-ene (TE) photopolymerizations.(6)Superbase photogenerator-based catalyzed thiol−acrylate Michael (TM) addition reaction and the combined system of TE and TM.(7)Dual-wavelength (UV and blue) controlled photopolymerization confinement (PC)(8)Dual-wavelength (UV and red) selectively controlled 3D printing.(9)Three-wavelength selectively controlled in 3D printing and additive manufacturing (AM).

With minimum mathematics, we present (for the first time) the synergic features and enhancing strategies for various systems of multi-components, co-initiators, co-monomers, and under one-, two-, and three-wavelength light. Therefore, this review provides not only the bridging between modeling and measurements, but also guidance for further experimental studies and new applications, based on the innovative concepts (kinetics/schemes) published in the most recent few years. 

## 2. Kinetic Systems and Discussions

### 2.1. Photo Crosslinking of Corneas 

Figure 1 shows the schematics of one-component photochemical pathways: (i) Radical-mediated and (ii) oxygen-mediated pathways. A typical example is applying a riboflavin solution (photoinitiation agent) to the cornea, which is irradiated by a UVA light (at 365 nm) for a procedure called corneal collagen crosslinking (CXL) [32,33,34]. Similar to the procedure of CXL is a type-II procedure for anti-cancer, in which cancer cells are killed by the oxygen singlet radical [35,36,37]. Synergic therapy combining photodynamic therapy (PDT) and photothermal therapy (PTT) has also been studied recently [38,39,40].

Greater details of Figure 1 are shown in Figure 2 for the kinetics of a photosensitizer (PS), monomer-A, with three reactive radicals, R’ and R and singlet-oxygen. The two pathways are described as follows. The ground state PS molecules (C) are excited by the UV light to its singlet excited state (C_1_), which could be relaxed to its ground state or to a triplet excited state (T*). In the type-I process, T* could interact directly with the substrate [A] to produce the first-radical (R’), which could produce (by chain reaction) a second-radical (R), which could interact with the ground state oxygen [O_2_], with the first-radical (R’), or the bimolecular termination (R^2^). For a type-II process, T* interacts with [O_2_] to form an oxygen singlet [^1^O_2_], which could relax to its ground state [O_2_], or interacts with the substrate [A]. For example, in a CXL procedure, A (monomer) is the corneal stroma matrix and C is riboflavin solution.

### 2.2. Synergic Effects of a Dual-Function Enhancer (3-Initiator System)

There are many strategies for improved photopolymerization such as the reduction of oxygen inhibition effects (OIH) and using co-initiators (or enhancers). A three-component system using phosphine to reduce the OIH effects during the free radical polymerization of (meth)acrylate monomers has been reported [17,18]. Figure 3 shows the kinetic scheme of a three-initiator system, [C], [A], and [B], in the presence of oxygen, using an enhancer-initiator [B]. Under near-infrared (NIR) light exposure, the initiator dye (C) is exited to its triplet-excited state, given by C*, which could react with the initiator [A] to produce an active radical (R) or react with the co-initiator [B]; where the dye [C] is regenerated in both reactions. The coupling of a radical [R] and oxygen [O_2_] produces a peroxyl radical [ROO°], which is too stable for the polymerization to proceed. Therefore, an enhancer-initiator [B] is required to create a less-stable radical [RO] for extra crosslinks of the monomer, [M]. We note that the initiator [B] plays a dual function of (i) regeneration of the dye [C], and (ii) reducing OIH and generating an extra active radical [RO] for improved conversion. Without the dual-function enhancer [B], OIH reduces the radical [R] and the conversion efficacy, in which an induction time is defined for the delayed rising of the conversion curve [18].

An example of the above system was reported by Bonardi et al. [17], in a three-component system of C/B/A, in which [C]= IR-140 borate, [B] = 4-(Diphenylphosphino) benzoic acid (4-dppba), and [A] = iodonium salt Ar_2_I + PF_6_^−^, with an initial concentration of [0.1/2.0/3.0] wt% and mixed in a monomer [M] = methacrylate. 

Several unique features for the conversion are demonstrated [18]. For example, reverse trends (roles) are found in (i) the light intensity and enhancer concentration, and (ii) the coupling rate constants of radical-oxygen and radical-monomer. The monomer conversion is an increasing function of enhancer, oxygen concentration, and light intensity. However, they have significantly different steady-state features. Lin et al. [17] reported that the steady-state conversion increases from 10% without the enhancer (with an enhancer concentration [B]_0_ = 0) to (30%, 50%, 80%) for [B]_0_ = (0.5, 1.0, 2.0)%.

### 2.3. Synergic Effects of a 3-Initiator Enhanced C/B/A System

Figure 4 shows the schematics of photochemical pathways in a three-initiator C/B/A system [20]. Figure 4 shows an example of the reported (CQ)/amine/AY system of Kirschner et al. [15], corresponding to our C/B/A system, where AY (aryliodonium ylides) is our [A] and the amine (our [B], the enhancer) could be Ethyl-4-(dimethylamino)benzoate (EDB) or 4-(dimethylamino)benzonitrile (DMABN) additives in multicomponent PI systems. The aryl radical (R) was generated through an electron transfer between CQ and the AY. In the presence of an amine (EDB or DMABN), additional reactions were expected [15] leading to additional initiating radicals (R), via the interaction of the excited molecule (CQ-H*) with AY. The free radical initiates the photopolymerization of the monomer, (meth-)acrylates, besides the oxygen-mediated photopolymerization. 

The measured system reported by Kirschner et al. [15] showed that the conversion could be improved by a higher concentration of the additive initiators and the kinetic rate constants. These features are consistent with our modeling. Competing mechanisms between the reduction of conversion due to the reduction of light intensity (in a thick polymer) and the reduction of oxygen inhibition (higher conversion) were analyzed in a thick polymer [20]. The optimal conditions are governed by the product function of the light intensity and main initiator concentration (C_0_), in which the conversion efficacy has a normal-trend proportional to C_0_I_0_, for the transient-state, but a reversed-trend for the steady-state. Strategies for an improved conversion include increasing the photoinitiator concentration, the light dose and intensity, the addition of oxygen scavengers, and the use of multiple photoinitiators. In the CQ/DMABN/AY system, Kirschner et al. [15] reported that higher AY (from 0% to 0.75%) leads to higher conversion (from about 40% to 60%).

### 2.4. Synergic Effects in a 3-Initiator (A/B/C) System for FRP and CP

Figure 5 shows the schematics of a three-initiator system, (A/B/C), with electron-transfer and oxygen-mediated energy-transfer pathways. A specific system was reported by Liu et al. [23], where [A] is the benzophenone (BP) photoinitiator, the co-initiator [B] is ethyl 4-(dimethylamino)benzoate (EDB), and [C] is (4-tert-butylphenyl)iodonium hexafluorophosphate (Iod). Under UV (365 nm) LED irradiation, [A] transforms from a ground state (PI) to an excited triple state ^1,3^PI. For BP alone, PI-H’ (or R) and PI(-H) (or R’) are the active species for FRP. In the presence of EDB, the extra radical PI-H’ is produced and could couple with [C] to produce aryl radical Ar’ and cation PI’, which lead to free radical (FRP) and cationic photopolymerization (CP), respectively. Associated with the photolysis of BPC1/Iod and BPC1/EDB/Iod, the photoredox catalytic cycle was proposed in a three-component PI/EDB/Iod system [23,24]. The regeneration of PI speeds up the photopolymerization and slows down the consumption of PI in the photolysis experiments. Trimethylolpropane triacrylate (TMPTA) and (3,4-epoxycyclohexane)methyl 3,4-epoxycyclohexylcarboxylate (EPOX) were used as benchmark monomers for FRP and CP, respectively. 

The co-initiators/additives B and C have dual functions of (i) regeneration of photoinitiator A and (ii) generation of extra radicals. The synergic effects lead to higher conversion of free radical polymerization (FRP) and cationic polymerization (CP), consistent with the measured work of Liu et al. [24]. However, there are other theoretically predicted new features (findings), which are either not identified or explored experimentally, including (i) co-initiator [C] always enhances both FRP and CP conversions, whereas co-initiator [B] leads to more efficient FRP, but it also reduces CP; (ii) the FRP conversion is proportional to the square-root of (bIg)([A] + [B] + [C]), whereas CP conversion is proportional to the linear power of (bIg)[A][C]/[B], where I is the light intensity, [A], [B], and [C] are the initial concentrations of the co-initiators, and b and g are rate constants; (iii) the dominant polymerization is FRP or CP depending on the relative concentration of [A], [C], and [B] and the rate constants that define the number of radicals; (iv) the steady state CP conversion profile is independent to the light intensity, whereas higher light intensity reaches a lower steady state value for the profile of FRP. The specific systems analyzed are benzophenone derivatives (A) ethyl 4-(dimethylamino)benzoate (B) and (4-tert-butylphenyl)iodonium hexafluorophosphate (C) under a UV (365 nm) LED irradiation; and two monomers of trimethylolpropane triacrylate (TMPTA, for FRP) and (3,4-epoxycyclohexane)methyl 3,4-epoxycyclohexylcarboxylate (EPOX, for CP).

We note that Figure 5 is more general than the Scheme proposed by Liu et al. [23], which ignored the coupling of PI^+^ and epoxy monomer producing a propagating cation (Q), which could be terminated by [B] as cationic polymerizations were not experimentally carried out in the presence of B due to its inhibitor effect. Furthermore, the measured data of Liu et al. [23] for the case of CP were limited to two initiators of [A] and [C], although three-initiator systems of [A]/[B]/[C] were studied in FRP. The modeled system of Chen et al. [24] in Figure 5 and the associated kinetic equations include three-initiator systems for both FRP and CP.

### 2.5. Copper-Complex (G1) Photoredox Catalyst in G1/Iod/NVK Systems 

Figure 6 shows the schematics of a three-initiator system, (A/B/N) for copper-complex (G1) photoredox catalyst systems for FRP and CP [28,29,30,31]. A specific measured system of G1/Iod/NVK related to Figure 6 was reported by Mokbel et al. [28] with a proposed scheme (shown by Figure 7), in which the G1 in combination with iodonium salt (Iod), (oxidizing agent) generates the radical species through an electron transfer reaction. A propagation system containing the N-vinylcarbazole (NVK) additive leads to simultaneous regeneration of G1 and the formation of highly reactive cations (Ph- NVK+), which can very efficiently initiate the CP conversion [28]. Figure 8 shows the 3D-photopolymerization experiments using an LED projector @405 nm [28].

The general conversion features of a three-initiator system (A/B/N), based on the proposed mechanism of Mokbel et al. [29,30], for both FRP of acrylates and the free radical promoted CP of epoxides using a copper complex as the initiator are summarized as follows based on the modeling of Lin et al. [30]. Higher FRP and CP conversion can be achieved by co-initiators concentration [B] and [N], via the dual function of (i) regeneration [A] and (ii) generation of extra radicals S’ and S. The FRP and CP conversion is proportional to, respectively, the nonlinear and linear power of bI[A][B], where b and I are the absorption coefficient and the light intensity, respectively. The system in the air has lower conversion than in laminate due to the oxygen-inhibition effects. For thick samples (with thickness z), there is an optimal concentration [A*], which is inversely proportional (bzI), in contrast with a very thin sample, in which the conversion is an increasing function of [A] and [B]. The unique feature of dark polymerization in CP conversion enables the polymerization to continue in living mode, in contrast with that of the radical-mediated pathway in most conventional FRP. 

### 2.6. Radical-Mediated Thiol-Ene (TE) Photopolymerizations 

Radical-mediated thiol-ene (TE) photopolymerizations, as shown by Figure 9, exhibit the advantages of being rapid and optically clear excellent mechanical properties, exhibit delayed gelation, are relatively uninhibited by oxygen, and enable radical polymerization of a wide range of thiol and vinyl functional group chemistries [14]. Depending on the specific ene selected, they exhibit reaction kinetics strongly dependent on the electronic density of the ene and the thiol-ene structures. However, competing vinyl homopropagation of the vinyl group, particularly for acrylates, is an undesirable side reaction in thiol−ene photopolymerizations [14].

Our numerical results for the conversion efficacy, C_T_ (for thiol [A]) and C_V_ (for ene [B]) show that the roles of the reaction rate ratio, R_K_ = k_P_/k_CT_, and the concentration ratio, R_C_ = [A]_0_/[B]_0_ are consistent with our predicted results based on analytic formulas, which provide more general features for the roles of R_K_ and R_C_, summarized as follows:(i)Without the viscosity or homopolymerization (or kC_V_ = 0) effects, [A] and [B] have an equal overall polymerization rate (R_P_); C_V_ (C_T_) is an increasing (decreasing) function of the ratio R_C_ = [A]_0_/[B]_0_. For R_K_ = 1 (or kp = k) C_T_, C_V_, and C_T_ have the same temporal profiles, but have a reversed dependence on R_C_.(ii)For R_K_ > >1, [A] and C_T_ are almost independent of R_C_, but the second-order correction is inversely proportional to R2, an opposite trend in comparing with C_V_. As predicted by analytic formulas.(iii)With the presence of the viscosity effect, the free volume is reduced when crosslink efficacy increases. The reduction factor only affects the propagation rate constant; therefore, the viscosity effect does not affect the efficacy for the case of R_K_ > > 1 and affects the efficacy for other ratios of kp and k C_T_ where the viscosity effect reduces the efficacy of [B].(iv)For an optically thick polymer, the influence of dynamic light intensity is due to PI depletion. In most previous modeling with constant light intensity, the assumption suffers an error of 5% to 20% (underestimated) for a crosslink depth (Z_C_) ranging 300 to 500 um.(v)Scaling law for the functional group concentration of thiol, [A], and ene, [B], given by [A]m[B]n. For R_K_ > >1, the polymerization rates are first order in the ene concentration (or n = 1.0) and nearly independent of the thiol concentration (or m = 0); in contrast, m = 1.0 and n = 0 for R_K_ < <1. For R_K_ values near unity, polymerization rates are approximately 0.5 order in both thiol and ene functional group concentrations (m = n = 0.5). However, a scaling law of m = 0.4 and n = 0.6 was found in an acrylate system (with R_K_ = 13), due to contributions from homopolymerization [14].

### 2.7. Superbase Thiol−Acrylate Michael (TM) Addition and TE/TM Systems 

As shown in Figure 10, the TM addition reaction offers high modulus materials for applications such as coatings, dental restorative materials, shape memory materials, and composites [12,13]. It also has the unique potential for long-term dark-cure capability and insensitivity to oxygen-inhibition effects. Claudino et al. [12] proposed a strategy of a TM addition reaction using a superbase photogenerator as the initiation system involving a photobase UV-initiator, such as 2-(2-nitrophenyl)propyloxy- carbonyl-1,1,3,3-tetramethylguanidine (NPPOC-TMG) and coumarin-TMG, which have very low basicity and remain relatively stable within formulated monomer mixtures, but once photocleaved, led to a dramatic increase in the basicity of the released organobase. Claudino et al. [12] also developed the modeling equations using a two-step reaction mechanism for the catalytic cycle and predicted the overall kinetic behavior, where the fundamental phenomena, driving mechanisms, and primary factors affecting TM are presented, but under certain assumptions. We plan to further improve the proposed kinetics of Claudino et al. [12] to include the vinyl group consumption by both propagation and the homopolymerization effect. Furthermore, the viscosity effect in the TE system [14] may also affect the conversion efficacy in the TM system. The light intensity, I(z,t), in the photoinitiation reaction was assumed to be time and spatially independent by Claudino et al. [12]. This assumption eliminates all the spatial profile information, and it is valid for optically thin samples and is limited to a small light dose. Greater detail about the viscosity effect was reported by Chen et al. [14] and Lin et al. [41]. Detailed discussion for the spatial profiles for thick samples was reported by Lin et al. [32]. 

Figure 11 shows the schematic for a combined TE and TM system under a dual-wavelength-initiated mixture of thiol ([T]), acrylates ([A]), and methacrylates ([M]) monomers. Notations used are PB for the photobase catalyst, B for the photochemically yielded active-site base; PI is the initiator for TM, with triplet excited state PT*; R’, R, S, and S’ are reactive species.

### 2.8. Dual-Wavelength (UV and Blue) Controlled Photopolymerization Confinement (PC)

A variety of photoresponsive materials such as conjugated polymers have been reported for PC in AM and more recently for 3D and 4D bioprinting [3,4,5]. Both spatial and temporal controlled 3D processes were reported using single- and multiple-wavelength lights. For 3D photo printings, two-stage polymerization under two wavelengths to eliminate the oxygen inhibition effects was also reported experimentally [9,10,11]. The advantages of dual-wavelength concurrent inhibition and initiation photopolymerization include (i) controllable high vertical print speeds, (ii) eliminating the need for thin, oxygen-permeable projection windows, (iii) single-step fabrication of cured materials, and (iv) rapid generation of personalized products. One additional advantage is that the reflow into the inhibition volume during printing can be optimized for large cross-sectional area parts.

Two different mechanisms of dual-wavelength, selectively controlled, photo-initiation and photo-inhabitation have been reported experimentally: (i) Oxygen inhibition reported by Childress et al. [10] and (ii) radical inhibition reported by de Beer et al. [8] and van der Laan et al. [9]. In the first mechanism, using red and UV light, the pre-irradiation time of the red light could be controlled to tailor the induction time, such that photosensitization and photoinitiation can be independently achieved for reduced oxygen inhibition for faster and more efficient UV-light polymerization. Figure 12 shows the schematics of photochemical dual-wavelength (blue and UV) controlled volumetric 3D printing and AM for parallel lights and orthogonal lights patterns [8,9]. In the second mechanism, as shown by a proposed scheme in Figure 13 [21], the photochemical decomposition of butyl nitrite results in the formation of nitric oxide ([N]), an efficient inhibitor of radical-mediated polymerizations, and alkoxide radical (X) for extra polymerization initiation, beside the reactive radical (R). Concurrent with the blue-light photo-orthogonal, patterned irradiation, the blue-light-produced initiation radical could be reduced/inhibited by [N], such that photopolymerization confinement (PC) is achieved [18]. For PC application, a large polymerization inhibition depth adjacent to the projection window and continuous part production at high translation speeds are desired.

As reported by van der Laan et al. [11], the effectiveness of a photoinhibitor is strongly monomer-dependent, which also requires: (i) A high conversion of blue-photoinitiation in the absence of the UV-active inhibitor; (ii) a strong chain termination with significant reduction of blue and UV conversion in the presence of a UV-active inhibitor; and (iii) short induction time or rapid elimination of the inhibitor species in the dark (or absence of UV-light), such that the initiation–inhibition cycles may be switched on and off rapidly. Fast switching time may be achieved by a high conversion rate or high blue-light intensity.

### 2.9. Dual-Wavelength (UV and Red) Controlled 3D Printing

There are many conventional strategies to reduce oxygen inhibition in photoinduced polymerizations. Physical methods include working in an inert or closed environment, increasing the photoinitiator concentration, increasing the light dose or light intensity (for reduced induction time), the use of multiple photoinitiators with different rates of initiation, or the addition of oxygen scavengers. Chemical mechanisms incorporate additives or suitably functionalized monomers, which are insensitive to oxygen, such as the TE and TM additive systems [12,14]. To overcome oxygen inhibition, phthalocyanines were explored as with a relatively long triplet state lifetime (5 to 350 micro-second) and a high quantum yield (0.58–0.65). 

Figure 14 shows the schematics of photochemistry for the red-light, oxygen-mediated (type-II) and UV-light, radical-mediated (type-I) pathways [19]. The strategy for the controlled initiation–inhibition switch is based on two mechanisms: (i) Oxygen-inhibition for improved conversion and (ii) radical-inhibition for spatial conformation in 3D printing. Figure 14 shows an example of the kinetics reported by Childress et al. [11], based on an ethyl ether acrylate (DEGEEA) mixed by zinc 2,9,16,23-tetra-tert-butyl-29H,31H-phthalocyanine (ZnTTP), with a distinct absorption peak at UV-365 nm and red-635 nm, such that it can be independently excited by a UV and red light, respectively.

### 2.10. Three-Wavelength Controlled in 3D Printing and Additive Manufacturing (AM)

Figure 15 shows the schematics of three photochemical pathways of a three-wavelength photopolymerization [22]: (i) The photoinitiator A (under blue light), (ii) B (under UV light), and (iii) oxygen-mediated C (under red light). A higher oxygen concentration leads to a lower conversion, which could be enhanced by reducing the S-inhibition via a red or blue light pre-irradiation. We found that pre-irradiation time is given by T_P_ = 200 s for red light only, and reduced to 150 s, for both red and blue light. The system under UV-only leads to a conversion lower than that of blue-only. However, conversion could be improved by the dual light (blue and UV), and further enhanced by the pre-irradiation of red-light. The two competing factors, N-inhibition and S-inhibition, could be independently and selectively tailored to achieve: (i) High conversion of blue-light (without UV-light), enhanced by red-light pre-irradiation for minimal S-inhibition; and (ii) efficient PC initiated by UV-light-produced N-inhibition for reduced confinement thickness and for high print speed.

The red-blue-UV system could be extended to the following as long as these three wavelengths have minimal overlap in their absorbance spectra, such as (i) red-light (635 nm), green (532 nm), and UV-A (365 nm); (ii) near-IR (750–810 nm), red (630–660 nm), and near UV (365–405 nm); where most of these lights are available from the output of LED and the associated photosensitizers (or photoinitiator). We note that one of the key features is that the three wavelengths must be separated without overlapping such that they can be orthogonally applied to the 3 initiators for independent control of the light.

## 3. Kinetics and Efficacy Formulas

We will first present the kinetic equations for more general systems than those selected systems in Section 2. We will then present (without detailed derivations) the efficacy formulas associated with the photopolymerization conversion and the key parameters for 3D printing and AM. Dynamic profiles produced numerically will be shown for selected systems with a comparison to measured data. Greater details and more complex formulas can be found in the References cited. 

### 3.1. The Kinetic Equations for General Systems

Photopolymerization, in general, includes free radical-mediated, cationic and anionic catalyzed, and atom transfer radical polymerization [1,2]. Two classes of photoinitiating systems were defined depending on the mechanisms of light conversion into chemical radicals [1]. In type I, a unimolecular reaction, the photoinitiator produces two radicals for free radical photopolymerization (FRP). A type-II system relies on the combination of two molecules; the first molecule absorbs the photon. It is the chromophore, often called the photoinitiator or the photosensitizer. The second one could be an electron donor or acceptor, or a hydrogen donor, the so-called coinitiator (or additive), which produces the initiating radicals (R) through an electron transfer when coupling with the excited triplet state of the initiator (T*) [1].

Type I photoinitiators can exhibit high quantum yields of radicals; however, their light absorptions are limited to the UV-to-blue region (360 to 430 nm) of the light spectrum. By contrast, type II systems with a combination of organic dyes and coinitiators provide tremendous flexibility in the selection of light wavelength from the UV to the near infra-red region (360 to 980 nm). However, these type-II, two-component systems have limited efficiency compared to type I systems [42]. 

In typical type II systems, photosensitizers with good absorption features in the UV-blue region include [42] benzophenones [43,44,45], thioxanthones [46,47,48], camphorquinone [49,50,51], and benzyls [52]. For visible light, photosensitizers can be selected from a whole panel of organic dyes, such as coumarins [53], xanthenic dyes [54,55,56], cyanine dyes [57], phenazine dyes, and pyrrome- thene dyes [57,58,59]. The hydrogen donor co-initiators are generally amines and thiols [60,61,62,63]. 

In photopolymerization, the monomer is converted to a polymer after the light irradiation of the photoinitiator (PI) or photosensitizer (PS). The UV (or visible or infrared) light-produced triplet excited state (T*) can couple with: (i) The monomer [M], (ii) the oxygen (if the system is in the air), or (iii) additives (or co-initiators) producing extra reactive radicals, which convert the monomer to a polymer. The chain growth of a polymer radical with m-links stops as a result of chain termination reactions. Termination of the radicals can occur due to self-recombination, radical-radical coupling, or reacting with the additives. Each radical becomes the center of the origin of a polymer chain. Kinetic equations of an m-component radical photopolymerization process may be described as follows. 

Considering an n-component system, with an initiator [A], its triplet excited state (T*), and n-additive (or co-initiators) Bn (n = 1,2,3), and one monomer, M (for FRP), the kinetic equations for each of the component concentration are given by [64,65]
(1)$$\frac{d\left[A\right]}{\mathrm{dt}}=-\left(\mathrm{bI}\left[A\right]-\mathrm{REG}\right)$$
(2)$$\frac{\mathrm{dBn}}{\mathrm{dt}}=-(k_{n}{\;B}_{n}+R_{n})T^{*}$$
(3)$$\frac{{\mathrm{dT}}^{*}}{\mathrm{dt}}=\mathrm{bI}\left[A\right]-T/g$$
(4)$$\frac{{\mathrm{dR}}_{n}}{\mathrm{dt}}=({\mathrm{kMT}}^{*}+{\;k}_{n}{\;B}_{n}T^{*}+\overset{\infty }{\underset{n=1}{\sum }}{\;k}_{n}B_{n}R_{n+1})-R_{n}/{g\text{′}}_{}$$
(5)$$\frac{\mathrm{dM}}{\mathrm{dt}}=-{\mathrm{kT}}^{*}M-\overset{\infty }{\underset{n=1}{\sum }}{\;K}_{n}R_{n}M\;$$
where I(z) is a light intensity; b is an affective absorption constant proportional to the light absorption and excited state quantum yield. Equation (2) shows the initiator excited triplet state (T), with a lifetime defined by g = 1/[k″ + kM + Sum), with Sum = summation of T and its coupling to all additives, k_n_B_n_T. Equation (4) defines its lifetime defined by 1/g′ = K_n_M + Sum′, with Sum′ = summation of its coupling to all radicals, k_n_R_n_R_m_. Equation (4) also shows the radical R_n_ produced from three terms, the type-I, unimolecule cleavage term, kMT*; the type-II bimolecular couplings of T* and additives (Bn), or enhanced extra radicals (Rn) from its coupling with Bn; and the radical-radical couplings, including a self-recombination (when n = n + 1). Rn. Equation (5) defines the monomer conversion total rate function given by the type-I term, kT*M, and the sum of all radical-mediated type-II terms, including contributions from additive-enhanced effects via the second term of Equation (4). A longer lifetime of T* or R (or large g and g′) leads to higher conversion efficacy. 

Solving for Equation (5), the conversion efficacy is defined by CE = 1 − M(t)/M_0_ for FRP, where M_0_ is the initial concentration of the monomers. RGE in Equation (1) is the initiator-regeneration term, which improves the conversion efficacy in a type-II system catalytic cycle. An example of a photoredox catalytic cycle of a three-component system, G1/Iod/EDB, is shown in Figure 6 [28]. The kinetic reactions for light initiated a copper complex (G1) to its triplet excited state (G1*), which couples with the radicals for FRP and CP conversation using radicals of Ar^o^ and EDB^o^ (for FRP), and radical EDB(+) for CP. 

### 3.2. Basic Formulas for Conversion and Rate Functions

Considering a simple system of Figure 1, a one-initiator system (in the air) with an electron-transfer pathway-1 and an oxygen-mediated energy-transfer pathway-2, the kinetic Equation (5) for the monomer of Figure 1 is given by [20]
(6)$$\frac{\mathrm{dM}}{\mathrm{dt}}=-({\mathrm{kT}}^{*}+\mathrm{KR}+K\prime \prime X)M$$
where R and X (singlet oxygen) are the reactive radicals produced from the coupling of T* with the monomer and with the oxygen, respectively. In a so-called quasi-steady-state condition for the radicals, we obtain, T* = bIC_0_/(kM), R = [bIC_0_/k′]^0.5^, and X = k″[O2], with b being an effective absorption constant proportional to extinction coefficient and the quantum yield of T*; I is the light intensity; C_0_ is the initial concentration of the initiator; [O2] is the oxygen concentration; and k′ and k″ are coupling constants for the coupling of T* and M, and T* and oxygen, respectively. 

We note that the rate function of Equation (6) has three terms: The direct type-I coupling term kT*M, the radical term KRM, and the singlet oxygen term, K″XM (oxygen-mediated).

Solving for Equation (6), the conversion efficacy is defined by CE = 1 − M(t)/M_0_, where M_0_ is the initial concentration of the monomer. We obtain [20] for the case of perfect regeneration (with dC/dt = 0), for type-I,
(7)CE=(1−F)−d′(1+F)
with F(t) = exp(-dt); with d = K(T_0_)^0.5^, d′= (k′T_0_)^0.5^/([O2]_0_K^2^), and T_0_ = bIC_0_. The CE (type-I) has a transient state, CE = dt − d′(2-dt) = (d + 2d′) t − 2d’; and steady-state CE = (1 − d′), a decreasing function of light intensity.

For type-II (from K″X term), we obtain
(8)CE=1−exp[−H(t)]
where H(t) = p′[1 − exp(-Dt)]/D, with p′ = (k_4_g′) (bIA_0_[O2]_0_), D = k_4_T_0_ + k″(T_0_/k)^0.5^. The CE (type-II) has a transient state CE = 1 − exp(-p′t) = p′t, but a steady state CE = p′/D, which is a decreasing function of light intensity.

### 3.3. Basic Formulas for 3D Printing 

For dual-wavelength (UV and blue) controlled photopolymerization confinement (PC), the maximum print speed (Smax) was defined by de Beer et al. [8], when the dose difference of blue light and UV light equals to a critical value (E*), and B_1_ = β B_2_, and we obtain a similar formula:(9)$$\mathrm{Smax}=\left[{\;B}_{20}-{\beta \;B}_{10}\right]/E^{*}$$
where B_j0_ = b_j_I_j0_(z,t), β = (C_2_b_20_/b_10_)/(gC_1_C_3_), Cj (with j = 1,2,3) are the concentration of the three co-initiators, I_j0_, (with j = 1,2) the UV and blue light intensity and b_j0_ are a coupling constant. The simplified function of de Beer et al. [9] is when β = b_20_/b_10_. 

Curing depth and inhibition zone are the critical parameters for 3D printing and AM. A curing depth (Z_C_) is defined by when the conversion efficacy is higher than a critical value (CR) or CE > CR. Using Equation (7), with ignored d′ = 0, we obtain
Zc = [1/(2.3a′C_0_]ln[K′E_0_^0.5^/lnE′)(10)
with K′ = K(bC_0_)^0.5^, E′ = 1/(1 − CR). The above curing depth (Z_C_) is proportional to the pillar height measured in AM [10].

### 3.4. Conversion Profiles 

We will show selected temporal profiles (numerically produced) for various systems as follows.

Figure 16 and Figure 17 show the dynamic profiles (in dual initiators system) of oxygen and the conversion at various light intensities, in which higher light intensity leads to faster oxygen depletion and higher efficacy [41]. Figure 18 shows conversion profiles in an enhancer system, C/B/A (as shown in Figure 4), in which a higher concentration [B]_0_ leads to faster and higher conversion [41]. Figure 19 shows the conversion profiles for the CQ/DMABN/AY system, in which a higher initiator concentration of CQ leads to higher conversion. Figure 20 and Figure 21 show the CP profiles of epoxy functions of the model resin in the air in the presence of curve-1 for 2-ITX/Iod (0.25/4.3 %*w*/*w*), curve-2 for Anthracene/Iod (0.23/4.8 %*w*/*w*), and curve-3 G1/Iod/NVK, demonstrating that G1 (copper complex) is the most efficient initiator [29,30]. Figure 22 shows the methacrylate conversion of a bisGMA/TEGMA resin formulated with 0.2 wt% CQ/0.5 wt% EDAB/0.5 wt% BN and subject to continuous exposure of blue light, but an on–off exposure of UV-light, serving as an optical switch [10,21]. Figure 23 shows that higher red-light pre-irradiation time or light dose leads to a lower induction time for a fixed red-light intensity, in 3D or AM systems. [10].

## 4. Conclusions

As shown in Table 1, we have reviewed the important topics that were recently reported or proposed to conduct. Various modeling and kinetic schemes are theoretically proposed and compared with specific reported measurements. We conclude the following important features:
(i)CXL using UVA (365 nm) and a riboflavin solution as the initiator (photosensitizer) has type-I and type-II FRP pathways. Oxygen plays an important role, especially for type-II, in which the oxygen singlet radical has been used to kill cancer cells.(ii)Synergic effects are achieved by a dual-function enhancer, in which the FRP is improved by the reduction of oxygen inhibition effects. The reported measurement system [17] is a three-component system of C/B/A, in which [C] = IR-140 borate, [B] = 4-(Diphenylphosphino) benzoic acid (4-dppba), and [A] = iodonium salt Ar_2_I+PF_6_^−^, with an initial concentration of [0.1/2.0/3.0] wt%, mixed in a monomer [M] = methacrylate. (iii)Synergic effects are achieved by a three-initiator system, with two pathways of electron-transfer and oxygen-mediated energy-transfer, in which the presence of amine produces additional initiating radicals and hence improves the FRP. The reported measurement system is a (CQ)/amine/AY system of Kirschner et al. [15], in which higher AY (from 0% to 0.75%) leads to a higher conversion (from about 40% to 60%).(iv)The reported measurement system [29] was the copper-complex (G1) photoredox catalyst in G1/Iod/NVK systems for FRP and CP, in which the co-initiators/additives Iod and NVK have dual functions of (i) the regeneration of the photoinitiator and (ii) the generation of extra radicals. The synergic effects lead to higher conversion of FRP and CP. (v)Radical-mediated thiol-ene (TE) photopolymerizations. It offers the advantages of being rapid and optically clear, exhibits delayed gelation, and is relatively uninhibited by oxygen for efficient FRP. (vi)Superbase photogenerator-based catalyzed thiol−acrylate Michael (TM) addition reaction. It has the unique potential for long-term dark-cure capability and insensitivity to oxygen-inhibition effects. A dual-wavelength combined system of TE and TM could offer very efficient conversion with controlled profiles.(vii)A dual-wavelength (UV and blue) system was reported for controlled photopolymerization confinement (PC) for volumetric 3D printing and AM using parallel lights and orthogonal lights patterns [11].(viii)A dual-wavelength (UV and red) selectively controlled 3D printing, in which red-light pre-irradiation improves the conversion [9,19]. (ix)A three-wavelength system is proposed for controlled 3D printing and AM, in which the two competing factors, N-inhibition and S-inhibition, could be independently and selectively tailored to achieve: (i) High conversion of blue-light (without UV-light), enhanced by red-light pre-irradiation for minimal S-inhibition; and (ii) efficient PC initiated by UV-light-produced N-inhibition for reduced confinement thickness and for high print speed.(x)For dual-wavelength (UV and blue) controlled photopolymerization confinement (PC), the maximum print speed (Smax) is proportional to the dose difference of blue light and UV light, shown by Eq. (9). Curing depth (Z_C_) is proportional to the light dose, as shown by Eq. (10), which also defines pillar height measured in AM.

To conclude, with minimum mathematics, we present (for the first time) the synergic features and enhancing strategies for various systems of multi-components, initiators, monomers, and under one-, two-, and three-wavelength light. Therefore, this review provides not only the bridging of modeling and measurements, but also guidance for further experimental studies and new applications in 3D printings and AM, based on the innovative concepts (kinetics/schemes). As a final remark, we note that the present review article focuses on the free-radical-mediated FRP and cationic-catalyzed CP, which have available experimental results and proposed schemes, as the basis of our kinetic modeling. Other processes involving 3D (and 4D) printings shall also include the reversible deactivation radical polymerization (RDRP) techniques such as nitroxide-mediated polymerization, (NMP) [66], atom transfer radical polymerization (ATRP) [67], and reversible addition–fragmentation chain transfer (RAFT) [68]. However, they are not the scope of the present article, and they can be found in recent review articles by Corrigan et al. [68] and Bagheri et al. [69].

## Figures and Tables

**Figure 1 polymers-13-02325-f001:**
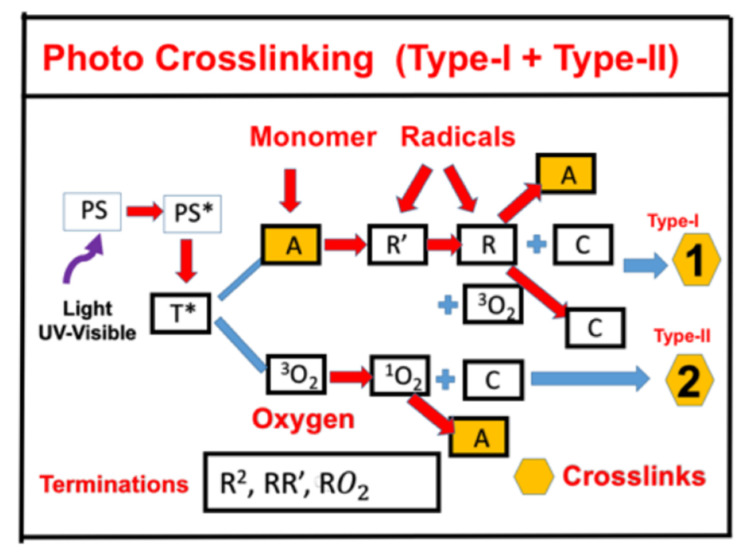
Schematics of photochemical pathways: (i) Radical-mediated (ii) and oxygen-mediated pathways; where PS is the ground state photosensitizer, having an excited state (PS*) and triplet state (T*), which interacts with the substrate A to form radicals R′ and R. It also may interact with the oxygen to form singlet oxygen. After Lin [33], Ophthalmology Research, 2017, 7, 1–8.

**Figure 2 polymers-13-02325-f002:**
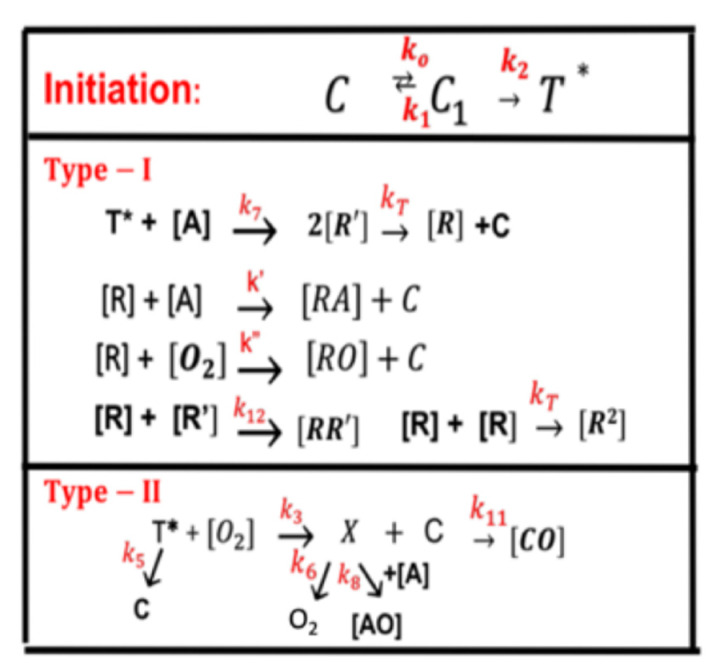
Schematics of photochemical pathways of a one-monomer system. The ground state PS (C) is excited by a UV-light to its singlet excited state (C_1_), which could be relaxed to its ground state or to a triplet excited state (T*). In the type-I process, T* could interact directly with the monomer [A] to generate a free radical (R′) by recombination. The radical R could interact with [A] for crosslinking, or oxygen [O_2_], or be terminated by coupling with R’, or bimolecular recombination (2R^2^). For the type-II process, T* interacts with [O_2_] to form an oxygen singlet [^1^O_2_] (X), which could relax to its oxygen [O_2_], or interacts with [A] for crosslinking, or coupling with C. All rate constants are shown in reds next to the arrows. After Lin [34], Ophthalmology Research, 2017, 7, 1–8.

**Figure 3 polymers-13-02325-f003:**
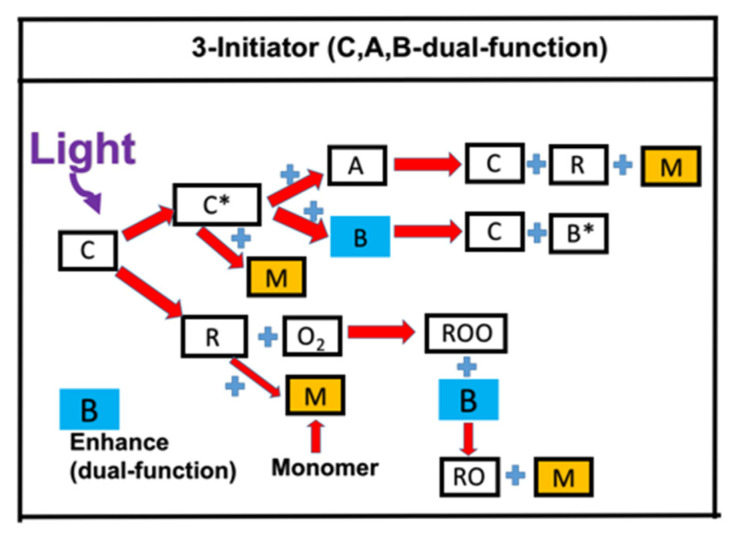
Schematics of photochemical mechanisms in a three-initiator (C/B/A) system under a near-infrared light; in which the initiator dye (C) is excited to its excited triplet state C*, which could react with [A] to regenerate [C] and produce active radical (R). R could initiate the crosslink of the monomer [M] or react with oxygen to produce a radical [ROO], which reacts with [B] to produce a radical [RO] causing an extra crosslink of [M]. After Chiu et al. [18], IEEE Access, 2020, 8, 83465–83471.

**Figure 4 polymers-13-02325-f004:**
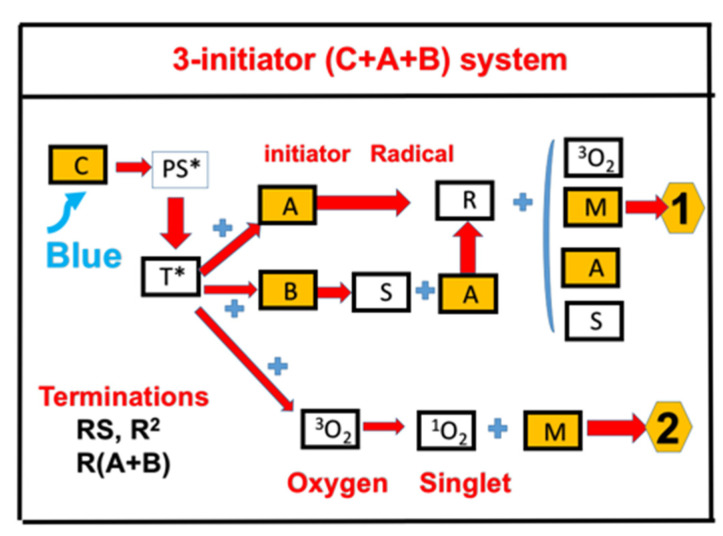
Schematics of photochemical pathways in a three-initiator C/B/A system, with the electron-transfer pathway-1, and the oxygen-mediated energy-transfer pathway-2. After Lin et al. [20], J Polymer Research, 2021, 28, 2.

**Figure 5 polymers-13-02325-f005:**
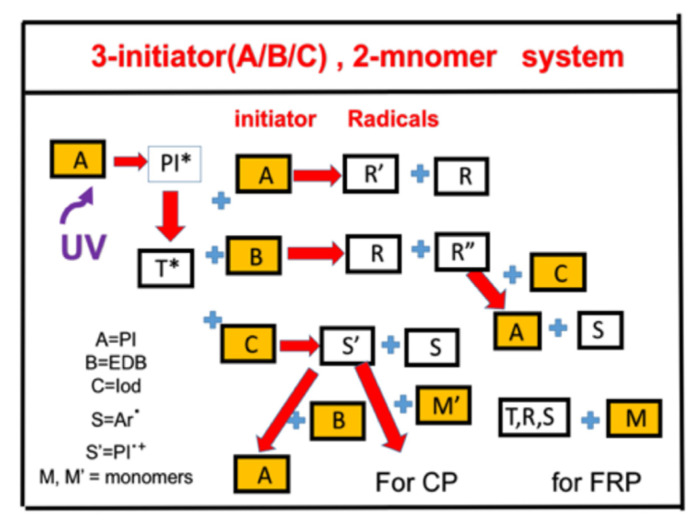
The schematics of a three-initiator system, (A/B/C), where A is the ground state initiator, with a first excited state PI*, and a triplet state T, which interacts with initiator [A] and [B] to produce radical R; and interacts with initiator [C] to produce radical S, in which the coupling of the radical R with [C] and S′ with [B] could lead to the regeneration of [A]. After Chen et al. [24], Res. Med. Eng. Sci. 2019, 8(2), 853–860.

**Figure 6 polymers-13-02325-f006:**
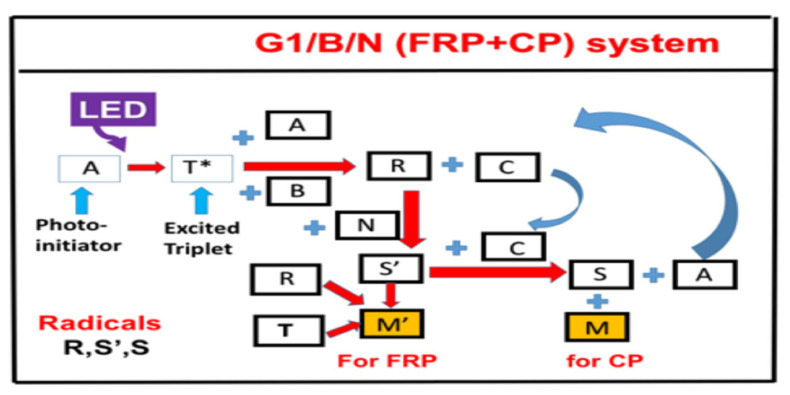
The schematics of a three-initiator system, (A/B/N), where A is the ground state of initiator-A, with an excited triplet state T, which interacts with co-initiator [B] to produce radical R and oxidized-A (or [C]); R interacts with co-initiator (or additive) N to produce radical S′, which couples with [C] to produce another radical S and lead to the regeneration of [A]. Monomer M′ and M coupled with radicals S′ and S for FRP and CP conversion, respectively. After Lin et al. [30], Polymers (2021, in press).

**Figure 7 polymers-13-02325-f007:**
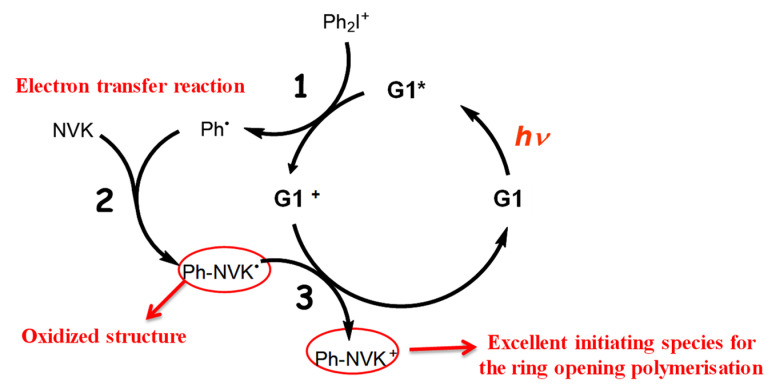
The proposed scheme for the G1/Iod/NVK copper-complex (G1) photoredox catalyst systems for FRP and CP reported by Mokbel et al. [29,30].

**Figure 8 polymers-13-02325-f008:**
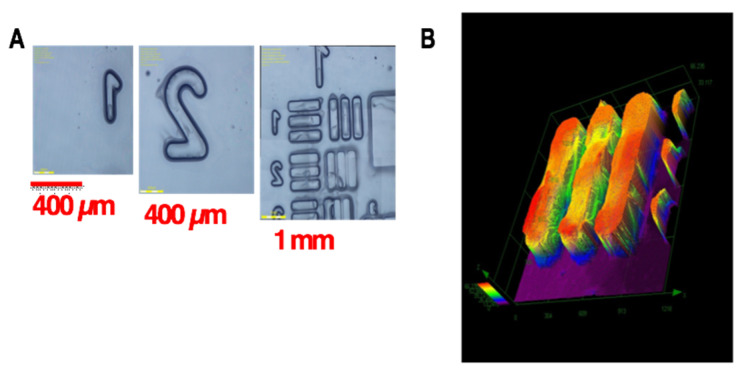
(**A**) 3D-photopolymerization experiments using an LED projector @405 nm where the numbers and patterns can be easily observed by a numerical microscop; (**B**) pattern characterized by profilometry [29].

**Figure 9 polymers-13-02325-f009:**
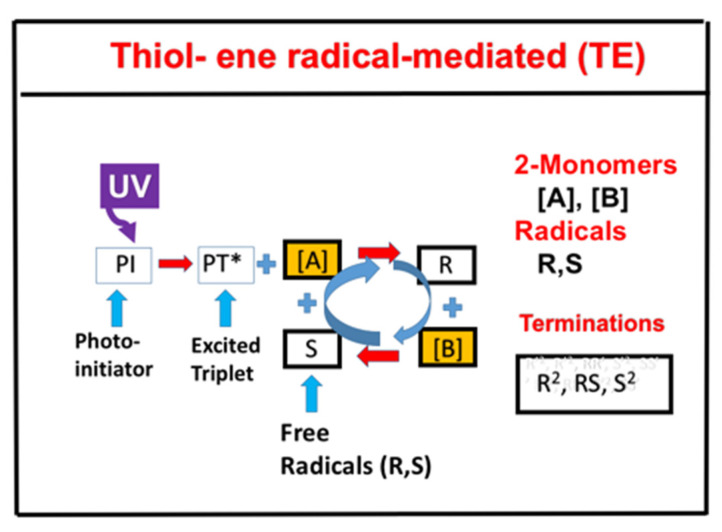
Schematics of thiol ([A]) and ene ([B]) functional groups, in which the thiyl radical R reacts with [B], to form a carbon radical (S) which reacts with thiol and regenerates R to form the reaction cycle; R and S could interact with each other or terminated by bimolecular recombination. After Chen et al. [14], Polymers 2019, 11, 1640; doi:10.3390/polym11101640.

**Figure 10 polymers-13-02325-f010:**
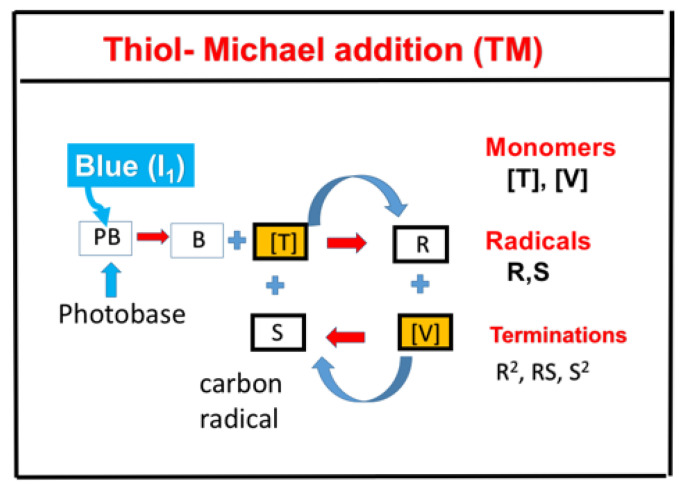
Kinetic scheme of a catalyzed thiol−Michael addition reaction with acid−base neutralization included as a termination mechanism, shown by B + [HA]; also included is the homopolymerization effect, shown by [C = C] + RC*. The reaction of deprotonations, propagation, and chain-transfer are given by B + [SH], [C = C] + RS*, and [SH] + RC*, respectively. Notations used are PB for the photobase catalyst, B for the photochemically yielded active-site base; [HB*], RS*, and RC* are intermediate reactive species for the conjugated acid, thiolate anion, and thiocarbanion, respectively. [HA] is the initial acidic impurities related to the induction time and limits the availability of base catalyst for deprotonation.

**Figure 11 polymers-13-02325-f011:**
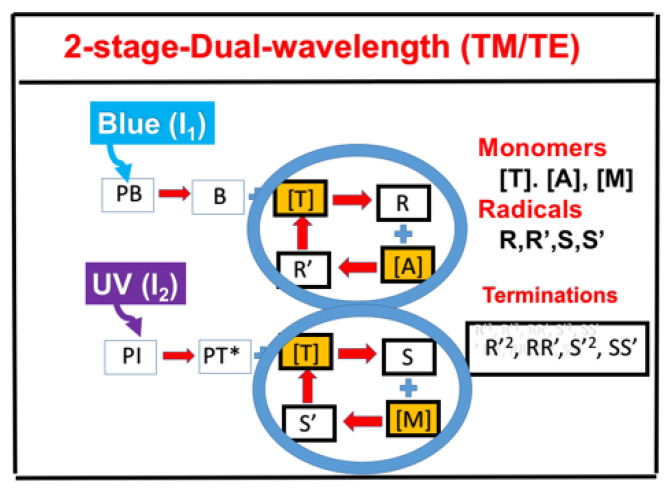
Kinetic scheme of a dual-wavelength-initiated mixture of thiol ([T]), acrylates ([A]), and methacrylates ([M]) monomers; TM for catalyzed thiol−acrylate Michael addition reaction; and TE for thiol-methacrylate reaction; notations used are PB for the photobase catalyst, B for the photochemically yielded active-site base; PI is the initiator for TM, with a triplet excited state PT*; R′, R, S, and S′ are reactive species. After Lin et al. (unpublished).

**Figure 12 polymers-13-02325-f012:**
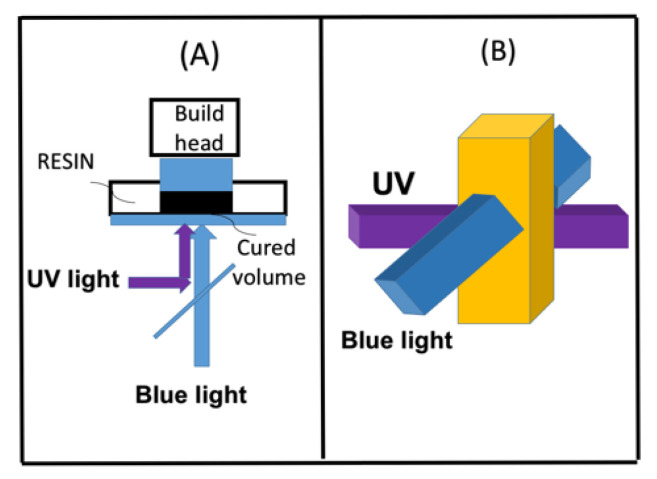
The schematics of photochemical, dual-wavelength (blue and UV) controlled volumetric 3D printing and additive manufacturing (AM) for parallel-light (**A**), and orthogonal-light patterns (**B**) [8,9].

**Figure 13 polymers-13-02325-f013:**
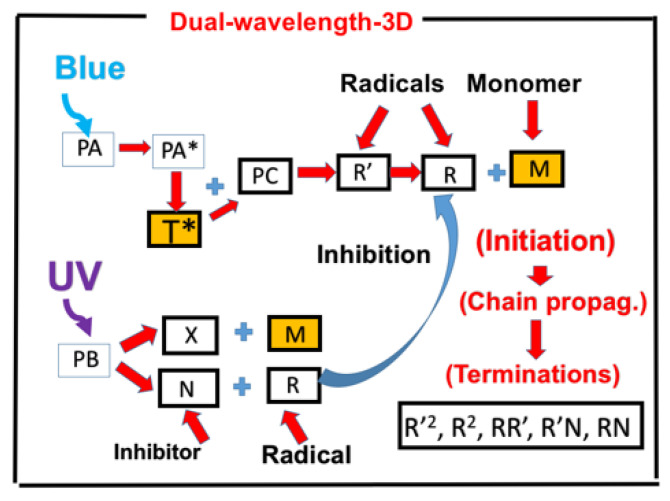
Schematics of photochemical pathways of dual-wavelength photopolymerization in which crosslinkers are formed via two pathways, via the photoinitiator PA (under a blue light) and PB (under a UV light). The initiation radicals R and [X] crosslink with the monomer [M]; whereas the inhibition radicals [N] reduce the conversion efficacy by reducing the active radicals (R′ and R). Shown also is the co-initiator (PC), which reacts with the triplet state of PA (T*) forming an intermediate radical (R′). Bimolecular termination of R′ produces a propagating radical (R), which leads to crosslinks; terminations could be also resulted by the interaction of R and R′, and R and [N]. After Lin et al. [21], Polymers, 2019, 11, 1819.

**Figure 14 polymers-13-02325-f014:**
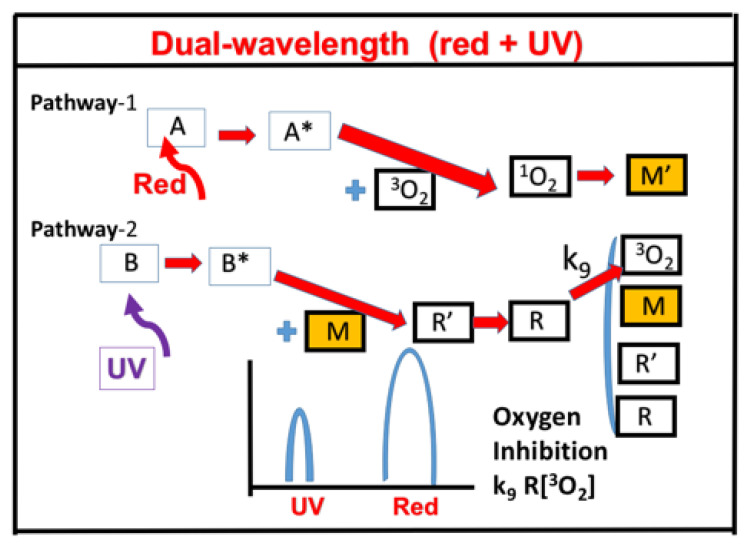
Schematics of photochemistry for pathway-1, red-light, oxygen-mediated (type-II) and pathway-2, UV-light, radical-mediated (type-I); where A and B are the ground state photosensitizer (PS) and photoinitiator (PI), with triplet excited states A* and B* and free radicals R′ and R; ^3^O_2_ and ^1^O_2_ are the ground state and singlet oxygen; M′ and M are the monomers. The absorption spectra are also shown with peaks at UV (365 nm) and red (635 nm). After Lin et al. [19], J Polymer Science, 2020, 58, 683–691.

**Figure 15 polymers-13-02325-f015:**
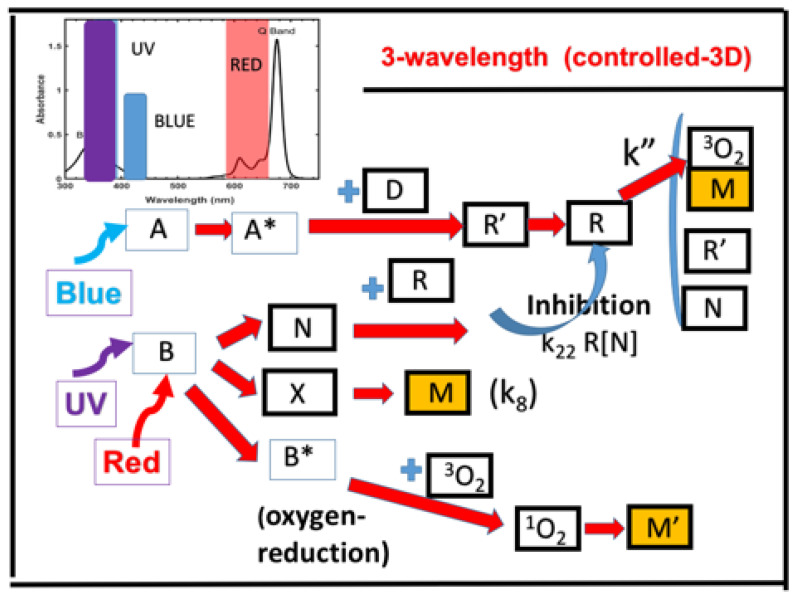
Schematics of photochemical pathways of a three-wavelength photopolymerization, in which crosslinkers are formed via three pathways: The photoinitiator A (under blue light), B (under UV light), and oxygen-mediated C (under red light). The initiation radicals R, [^1^O_2_], and [X] initiate the monomer [M] polymerization, whereas the inhibition radical [N] reduces the active radical R. The co-initiator (D) is also shown, which reacts with the triplet state of A (A*) forming an intermediate radical (R′) and a reactive radical (R), initiating crosslinkers; terminations may be resulted by the interaction among R′, R, and [N]. We note that (as shown in the upper left corner), the three wavelengths are orthogonally applied to the 3 initiators. After Lin et al. [22], IEEE Access, 2020, 8, 49353–49362.

**Figure 16 polymers-13-02325-f016:**
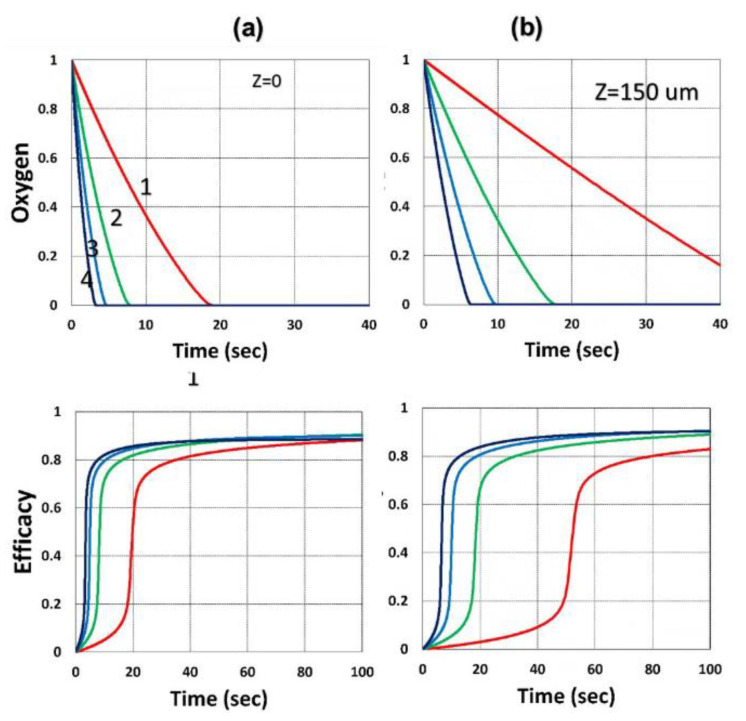
Dynamic profiles of oxygen and the conversion at various light intensities I_0_ = (0.5,2,5,10) mW/cm^2^, for curves (1,2,3,4); at z=0 ((**a**), on surface) and 150 um (**b**) [41].

**Figure 17 polymers-13-02325-f017:**
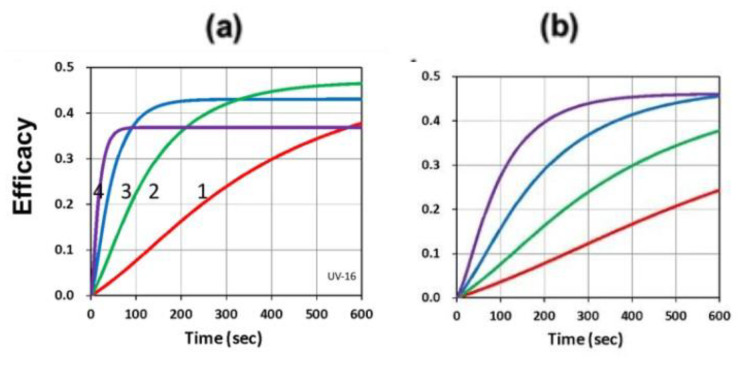
UV-light-initiated conversion efficacy for (**a**) b_2_ = (0.05, 0.08, 0.11, 0.14), for curve (1,2,3,4); and for (**b**) I_20_ = (10,20,40,80) mW/cm^2^. [41].

**Figure 18 polymers-13-02325-f018:**
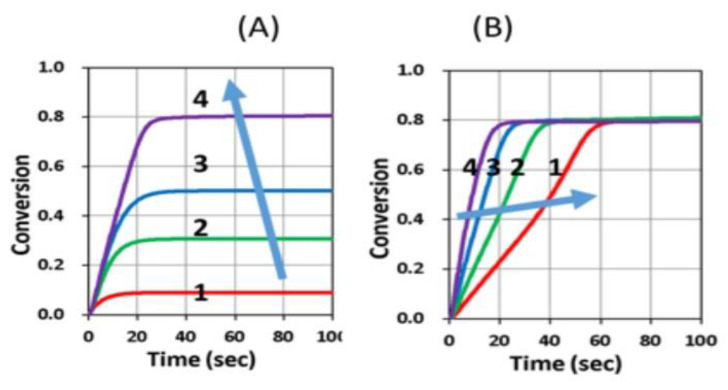
Conversion profiles in an enhancer system: (**A**) Concentration [B]_0_ = (0,0.5,1.0,2.0) %, for curve (1,2,3,4); with [A]_0_ = 3.0%, [C]_0_ = 0.1%, b′= bI_0_ = 0.6 (1/s/%); and (**B**) coupling constant b′ = bI_0_ = (0.15,0.3,0.6,1.2) (1/s/%), with [A]_0_ = 3.0%, [B]_0_ = 2.0%, [C]_0_ = 0.1%, [O_2_]_0_ = 1.5mg/L; and k′ = 0.7, k″ = 2.5, k_T_ = k_2_ = 2.0, k_1_ = 0.7, k_3_ = 8, k_5_ = k_7_ = 1 (1/s). After Chui et al. [18].

**Figure 19 polymers-13-02325-f019:**
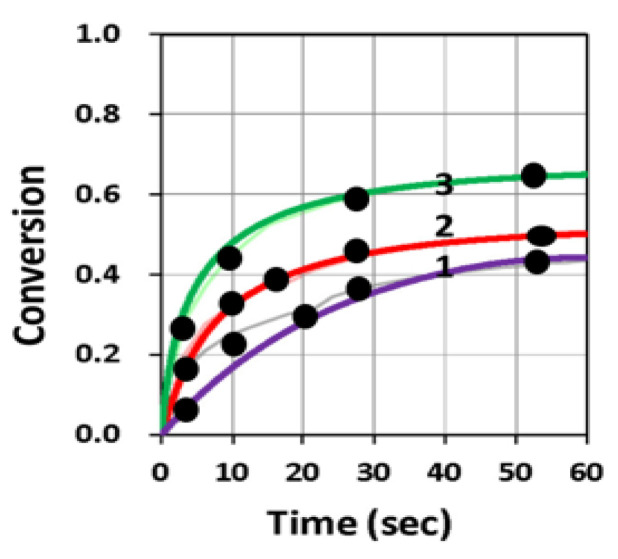
Conversion profiles for the CQ/DMABN/AY (1.5/0.6/0.75 %*w*/*w*) system; curve-1 without AY (or A_0_ = 0), and curve 2 and 3 for A_0_ = 0.2 and 0.75% with B_0_ = 0.6% and C_0_ = 1.5%. Dots are measured data of Kirschner et al. [15], and solid curves are fit modeling curves of Lin et al. [20].

**Figure 20 polymers-13-02325-f020:**
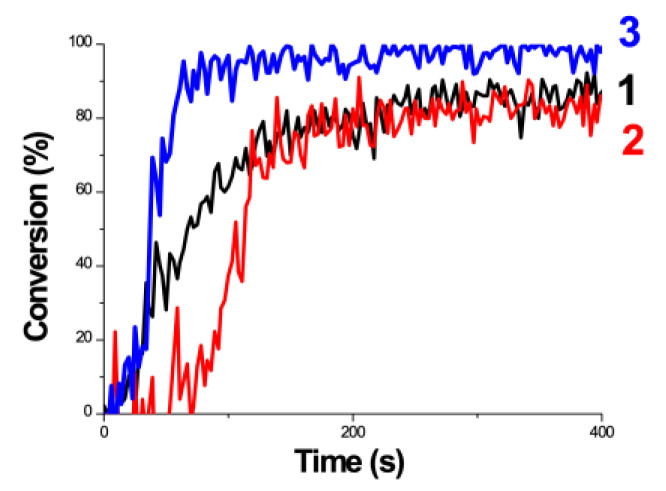
Photopolymerization profiles of epoxy functions of the model resin in the air in the presence of (1) 2-ITX/Iod (0.25/4.3 %*w*/*w*), (2) Anthracene/Iod (0.23/4.8 %*w*/*w*), and (3) G1/Iod/NVK (0.22/5/1 %*w*/*w*/*w*), upon LED@405 nm exposure, sample thickness = 1.4 mm. After Mokbel et al. [29].

**Figure 21 polymers-13-02325-f021:**
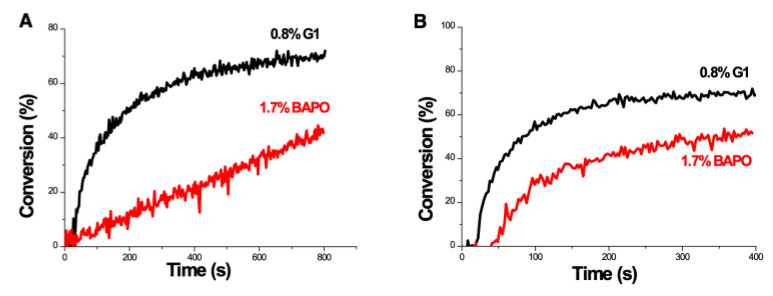
Photopolymerization profiles (epoxy function conversion vs. irradiation time) for DGEBA in the air in the system of: (**A**) G1/Iod/NVK (0.8/4.6/1 %*w*/*w*/*w*), or (**B**) BAPO/Iod/NVK. After Mokbel et al. [29].

**Figure 22 polymers-13-02325-f022:**
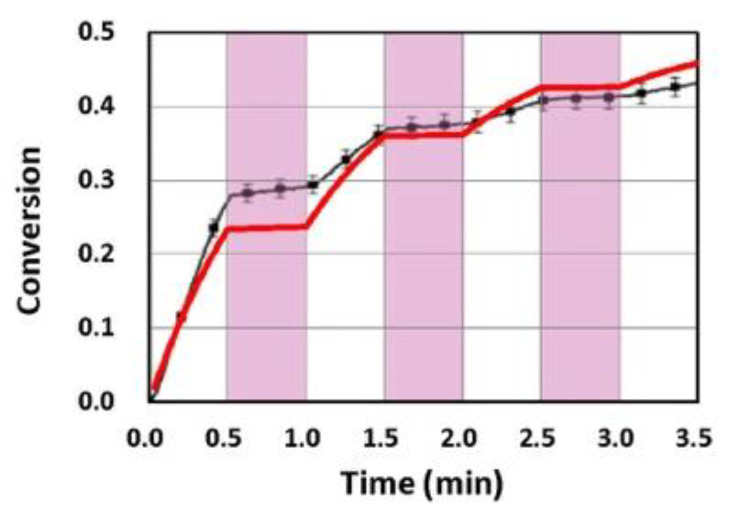
Methacrylate conversion of a bisGMA/TEGMA resin formulated with 0.2 wt% CQ/0.5 wt% EDAB/0.5 wt% BN and subject to a continuous exposure of blue light, but an on–off exposure of a UV-light for 0.5 min, as indicated by the violet vertical areas; where black bars are measured data from van der Laan et al. [10] and the red curve is the theoretical simulation of Lin et al. [21].

**Figure 23 polymers-13-02325-f023:**
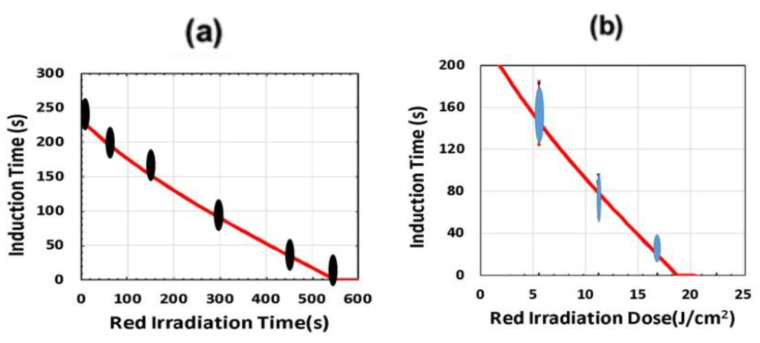
Induction time (T_ID_) versus red-light pre-irradiation time (T_P_) (**a**) for a fixed I_10_ (red) = 34 mW/cm^2^ and I_20_(UV) = 10 mW/cm^2^; and (**b**) for T_ID_ vs. the red-light dose for I_10_ (red) = (10,20,30) mW/cm^2^. Red curves are modeling data by Lin et al. [19]; also shown are the measured data (in bars) of Childress et al. [10].

**Table 1 polymers-13-02325-t001:** Summary of enhancing strategies for photopolymerization.

System	Light	Enhancer	References
One-component		co-initiators	
	blue (477 nm)	CQ/EDB/AY	Kirschner et al. [15]
	UV (365 nm)	BP/EDB/Iod	Liu et al. [23]
	green (532 nm)	CQ/rose-Bengal	Wertheimer et al. [31]
	NIR (785 nm)	phosphine/Iod	Bonardi et al. [17]; Chiu et al. [18]
two-component		co-monomers	
	UV (365 nm)	thiol–Vinyl (Michael) -	Claudino et al. [12]
		thiol–Ene	Chen et al. [14]
	dual (365 + 660 nm)	-DEGEEA/ZnTTP	van der Laan et al. [9]
			Childress [10]
			Lin et al. [19]
	Dual (365 + 430 nm)	-DEGEEA/ZnTTP	Scott et al. [11]
		-	Lin et al. [20,21]
	3-wave (365,430, 660)	DEGEEA/ZnTTP	Lin et al. [[20],[21]
three-component	UV (365 nm)	Thiol BMP/EVS/BA	Huang et al. [13]
	UV (365 nm)	PI/EDB/Iod	Liu et al. [24]
	UV (405 nm)	Meth/Iod/NPG	Abdallah et al. [26,27]
	UV (405 nm)	G1/Iod/NVK	Mokbel et al. [29,30]
	5.9	352.7	Lin et al. [31]

* CQ = Camphorquinone, AY = aryliodonium ylides, EDB = ethyl 4-(dimethylamino)benzoate; Iod= (4-*tert*-butylphenyl)iodonium hexafluorophosphate; BP = benzophenone; DEGEEA = ethyl ether acrylate, ZnTTP = zinc 2,9,16,23-tetra-tert-butyl-29H,31H-phthalocyanine; BMP = 1-butyl mercaptopropionate; EVS = ethyl vinyl sulfone; BA = 1-butyl acrylate. G1 = copper complex, NVK = N-vinylcarbazole.

## Data Availability

May be found from the cited references.

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
