# Peer review of "A Critical Review for Synergic Kinetics and Strategies for Enhanced Photopolymerizations for 3D-Printing and Additive Manufacturing"

_polymers, 2021, doi:10.3390/polym13142325_

Round 1

Reviewer 1 Report

The work „Synergic kinetics and strategies for enhanced photopolymerizations for 3D-printing and additive manufacturing Critical Review”  presented me to review for Polymers journal is very interesting and associated with the latest trends in photopolymerization, i.e. 3D printing. Authors on the basis of many significant works in this field made a critical review and give few kinetic schemes for several reaction mechanisms. There are many editing errors in this manuscript, making it difficult to read: like missing brackets, different fonts, typos, lack of spaces, unnecessary words, etc., for example:

Page 11, line 373: “the viscosity effect affects does not affect the efficacy for the case” or line 377: “light intnesity”

Page 6 line 237: “The aryl radicals R) were”

Page 16 line: 538: “terminationand in thenpresentce”

The work should be carefully read and corrected.

There are also some minor remarks:

  • On Figure 2 - Type II, after arrow k6 should be [O2] not [Q2]
  • Figures: One time for excited triplet authors used T (Figure 6) and for another time T* (Figure 2). Shouldn't they be labeled with the same symbol?
  • On page 17 line 548 literature reference should be [8] not [9]
  • Section 2.6, page 10, line 364, F should be explain
  • All symbols used in section 3 should be explain to facilitate understanding of the derived equations. Moreover authors forget to use subscripts for the appropriate parameters (for instance RP, RK, kCT)
  • Page 11 line 400, authors refer to unpublished work. I think that authors should just refer to their works. Instead of writing:"Lin et al. (unpublished) further improve the proposed kinetics", they can write our investigations further improve the proposed kinetics, or something like that. This statement “Lin et al. (unpublished)” appears few times in this work.
  • I think that radical should have dot in symbol R, not the R alone, then the reader will know on the first sight that this is a radical not the rate of reaction or some other constant in the equation/mechanism/reaction

Author Response

(1) Page 11, line 373: “the viscosity effect affects does not affect the efficacy for the case” or line 377: “light intnesity”   REPLY: spelling error fixed.

(2) Page 6 line 237: “The aryl radicals R) were”    

      REPLY: fixed as: ....."radical (R) was"

(3) Page 16 line: 538: “terminationand in thenpresentce”,  REPLY: it  was      corrected as  " with bimolecular termination and in the presence of " 

(4) On Figure 2 - Type II, after arrow k6 should be [O2] not [Q2]  REPLY: yes, agreed and fixed. ï‚·    

(5) Figures: One time for excited triplet authors used T (Figure 6) and for another time T* (Figure 2). Shouldn't they be labeled with the same symbol?    ï‚·      REPLY: yes, agreed and we have used  T* in ALL revised Figures, 2 to 6. 

(6) On page 17 line 548 literature reference should be [8] not [9]   REPLY: fixed

(7) Section 2.6, page 10, line 364, F should be explained.  

   ï‚·  REPLY: it was removed, because it is not needed in the article. (it was used in our original article [14], which defined F mathematically).

(8) All symbols used in section 3 should be explain to facilitate understanding of the derived equations.

REPLY: yes, we have expanded Section 3 and added more text and equations, accordingly. However, this might not consistent with what we claimed of "minimum mathematics in this Review article". 

(9) authors forget to use subscripts for the appropriate parameters (for instance RP, RK, kCT)   REPLY: fixed. 

(10) Page 11 line 400, authors refer to unpublished work. I think that authors should just refer to their works. Instead of writing:"Lin et al. (unpublished) further improve the proposed kinetics", they can write our investigations further improve the proposed kinetics, or something like that. This statement “Lin et al. (unpublished)” appears few times in this work.

ï‚·   REPLY: agreed, and we have revised the related text as follow: We plan to further improve the proposed kinetics of Claudino et al [12] to include the vinyl group consumption by both propagation and the homopolymerization effect. Furthermore, the viscosity effect in the TE system [14] may also affect the the conversion efficacy in the TM system. The light intensity, I(z,t), in the photoinitiation reaction was assumed as time and spatially independent by Claudino et al [12]. This assumption eliminates all the spatial profile information, and it is valid for an optically-thin samples and limited to small light dose. Greater detail about viscosity effect was reported by Chen et al [14] and Lin et al [41]. Detailed discussion for the spatial profiles for thick samples was reported by Lin et al [32]. 

(11)  I think that radical should have dot in symbol R, not the R alone, then the reader will know on the first sight that this is a radical not the rate of reaction or some other constant in the equation/mechanism/reaction.  

 ï‚·    REPLY: yes, we totally agreed with the suggestion.   However, changing R to R0 will involve too many corrections in the whole article text and most of the Figures, This might be a tough job for us and might also produce more errors (in case we miss some places having "R"). Therefore, if it is ok for the Reviewer, we prefer to keep R as it is. In fact, in our original published articles, the "R" was well defined in our kinetic equations. And in most text, we said radical (R) rather than just "R", to remind the readers.   ï‚·    PS: in most of our Figs and text, we used symbols of R,S,R',S'  for various "reactive radicals", if we would need to change them to Ro, So, Ro', So',  The recombination of R, S and their coupling also involve with R2, S2, RR', RS  etc, which are too complex to use Ro2 etc.  ï‚·     

Reviewer 2 Report

The proposed paper describes a review on synergic features and enhancing strategies of photopolymerization for 3D-printing and additive manufacturing. Overall the paper covers a few very specific issues and does not provide a broader perspective and comparison with other methods. The paper covers only 41 references, which may be fine for a regular article, but is not enough for a proper and thorough review. There are 15 self-citations of the main Author, which is 37% of all of the references - a relatively high percentage even for a review. In my opininon Authors should expand the proposed paper and add more references to other studies from the same field, providing broader perspective on the presented issues.

Author Response

ï‚·    REPLY: yes, we totally agreed with the good suggestions. Accordingly, we have expended Section 3 for more background on the materials aspects (from other studies) and adding about 40 more references. We also added more mathematics in Section 3. However, this is conflicting to our claimed "minimum mathematics in this Review article" (which is written in a format mainly for experimental readers)  

Round 2

Reviewer 2 Report

Still, there is a lot of self-citations in the paper, but if it is not a concern for the Editor, then I suggest aceptance of the manuscript.